# Systematics of Ditaxinae and Related Lineages within the Subfamily Acalyphoideae (Euphorbiaceae) Based on Molecular Phylogenetics

**DOI:** 10.3390/biology12020173

**Published:** 2023-01-21

**Authors:** Josimar Külkamp, Ricarda Riina, Yocupitzia Ramírez-Amezcua, João R. V. Iganci, Inês Cordeiro, Raquel González-Páramo, Sabina Irene Lara-Cabrera, José Fernando A. Baumgratz

**Affiliations:** 1Programa de Pós-graduação em Botânica, Escola Nacional de Botânica Tropical, Instituto de Pesquisas Jardim Botânico do Rio de Janeiro. Rua Pacheco Leão 2040, Rio de Janeiro 22460-030, RJ, Brazil; 2Real Jardín Botánico (RJB), CSIC, Plaza de Murillo 2, 28014 Madrid, Spain; 3Laboratorio de Sistemática Molecular, Administración Santa María, Facultad de Biología, Universidad Michoacana de San Nicolás de Hidalgo, Morelia Michoacán 58091, Mexico; 4Programa de Pós-Graduação em Botânica, Universidade Federal do Rio Grande do Sul, Av. Bento Gonçalves 9500, Porto Alegre 91501-970, RS, Brazil; 5Programa de Pós-Graduação em Fisiologia Vegetal, Departamento de Botânica, Campus Capão do Leão, Universidade Federal de Pelotas, Caixa Postal 354, Pelotas 96010-900, RS, Brazil; 6Núcleo de Conservação da Biodiversidade, Instituto de Pesquisas Ambientais, Av. Miguel Stefano 3687, São Paulo 04301-012, SP, Brazil

**Keywords:** Adelieae, *Argythamnia*, *Caperonia*, Caperonieae, *Chiropetalum*, Ditaxeae, *Ditaxis*, phylogenetics, *Philyra*

## Abstract

**Simple Summary:**

This study represents the most comprehensive phylogenetic reconstruction of the plant subtribe Ditaxinae and related taxa within Acalyphoideae (Euphorbiaceae). The taxonomy of this group, mainly based in morphology, has long been controversial. Here, we present a new taxonomic classification at the genus and tribe ranks using a solid phylogenetic framework. We also provide key morphological synapomorphies supporting the main recovered clades.

**Abstract:**

The subtribe Ditaxinae in the plant family Euphorbiaceae is composed of five genera (*Argythamnia*, *Caperonia*, *Chiropetalum*, *Ditaxis* and *Philyra*) and approximately 120 species of perennial herbs (rarely annual) to treelets. The subtribe is distributed throughout the Americas, with the exception of *Caperonia*, which also occurs in tropical Africa and Madagascar. Under the current classification, Ditaxinae includes genera with a questionable morphology-based taxonomy, especially *Argythamnia*, *Chiropetalum* and *Ditaxis*. Moreover, phylogenetic relationships among genera are largely unexplored, with previous works sampling <10% of taxa, showing Ditaxinae as paraphyletic. In this study, we inferred the phylogenetic relationships within Ditaxinae and related taxa using a dataset of nuclear (ETS, ITS) and plastid (*pet*D, *trn*LF, *trn*TL) DNA sequences and a wide taxon sampling (60%). We confirmed the paraphyly of Ditaxinae and *Ditaxis*, both with high support. Following our phylogenetic results, we combined *Ditaxis* in *Argythamnia* and upgraded Ditaxinae to the tribe level (Ditaxeae). We also established and described the tribe Caperonieae based on *Caperonia*, and transferred *Philyra* to the tribe Adelieae, along with *Adelia*, *Garciadelia*, *Lasiocroton* and *Leucocroton*. Finally, we discuss the main morphological synapomorphies for the genera and tribes and provide a taxonomic treatment, including all species recognized under each genus.

## 1. Introduction

The systematics of Euphorbiaceae Juss. have undergone substantial changes in the last two decades stemming from studies in molecular systematics. The family is currently classified into four subfamilies (Acalyphoideae Beilschmied, Cheilosoideae K.Wurdack & Petra Hoffm., Crotonoideae Beilschmied and Euphorbioideae) [1,2,3,4,5]. Phylogenetic studies have led to updates in the systematics of Euphorbiaceae, where two biovulate subfamilies were segregated and elevated to the family level, Phyllanthaceae Martinov, Picrodendraceae Small and Putranjivaceae Endl. [2,6]. Acalyphoideae was recognized as a subfamily in 1975 [7] and currently comprises 14 tribes, 23 subtribes, 99 genera and approximately 1860 species [5,8]. The subfamily is distributed worldwide, except in polar regions, with greater diversity in tropical and subtropical areas [5,9,10,11].

In the classification proposed by Webster [5], the tribe Chrozophoreae (Müll.Arg.) Pax & K.Hoffm was composed of the subtribes Ditaxinae Griseb., Speranskiinae G.L.Webster and Chrozophorinae (Müll.Arg.) Pax & K.Hoffm. Ditaxinae was proposed in 1859 [12], with the genera *Argythamnia* P.Browne (=*Chiropetalum* A.Juss.), *Caperonia* A.St.-Hil. and *Ditaxis* Vahl ex A.Juss. Later, Müller [13] presented a classification for the tribe Acalypheae that consisted of 11 subtribes, including Chrozophorinae Müll.Arg. (containing *Argythamnia*, *Chiropetalum*, *Ditaxis* and *Philyra* Klotzsch) and Caperoniinae Müll.Arg. (including only *Caperonia*). Müller differentiated these subtribes based on the staminate flowers having a rudimentary ovary present at the apex of the staminal column in *Caperonia* and absent in the other genera.

In 1912, a new classification system called “Chrozophorinarum’’ was put forward by Pax and Hoffmann, wherein Ditaxinae was treated as a synonym of Chrozophorinae-regularis, which had been circumscribed with the genera *Aonikena* Speg., *Argythamnia*, *Caperonia*, *Chiropetalum*, *Chrozophora* Neck. ex A.Juss., *Ditaxis* and *Philyra* [14]. Webster re-established Ditaxinae to include *Argythamnia*, *Caperonia*, *Chiropetalum* (= *Aonikena*), *Ditaxis* and *Philyra* [7].

In its current circumscription, Ditaxinae consists of five genera (*Argythamnia*, *Caperonia*, *Chiropetalum*, *Ditaxis* and *Philyra*) and around 120 species of herbs, subshrubs, shrubs, and small trees, widely distributed in the New World (all genera) [5,15,16,17,18,19,20,21] and continental Africa and Madagascar (only *Caperonia*) [5,17,21]. 

*Caperonia* has approximately 35 herbaceous and subshrub species, of which 29 are distributed in the New World and seven in Africa/Madagascar. In the New World, it occurs from Mexico to central Argentina, with one species introduced in the southern United States. *Caperonia* has its greatest diversity in tropical and subtropical regions, and is the only genus of the tribe occurring in the Amazonian region, exclusively in marshy environments [5,14,17,21]. *Argythamnia* is composed of 19 species of perennial herbs, subshrubs to shrubs distributed in Central America (Caribbean and Mexico), where it is restricted to seasonally dry tropical forests and coastal vegetation [5,22]. *Chiropetalum* consists of 21 species of herbs and subshrubs and is disjunct between Mexico (two species) and South America (19 species), where it occurs from Peru to Patagonia, with its highest diversity in northern Argentina and southern Brazil [15,20,23]. *Chiropetalum* occurs in a variety of habitats, including a range of dry and humid forests, arid environments, grasslands and coastal vegetation. *Ditaxis* is the most species-rich genus of the subtribe with approximately 45 species, ranging from herbs to shrubs, and is widely distributed from the southern United States to northern Patagonia, in Argentina [18,24,25,26]. *Ditaxis* occupies various habitats, such as deserts and grasslands, but most species occur in seasonally dry tropical forests [18,25,26]. Finally, the monotypic genus *Philyra* (*P. brasiliensis* Klotzsch) is a shrub or a small tree. The genus is restricted to central and eastern South America, growing exclusively in seasonally dry tropical forests [5,19].

*Argythamnia*, *Chiropetalum* and *Ditaxis* form a group of great morphological complexity that has undergone many taxonomic changes. However, few studies have approached the three genera all together to understand their phylogenetic relationships [13,14,22,23,27]. *Argythamnia*, *Chiropetalum* and *Ditaxis* have sometimes been treated as subgenera of *Argythamnia s.l.* [22,23,24,27]. Currently, these taxa are treated at the genus level, but there is still disagreement among taxonomists. Recent studies, using DNA sequence data, have attempted to resolve the relationship among *Argythamnia*, *Chiropetalum* and *Ditaxis* [15,28], but their phylogenetic analyses revealed topologies with low support in some clades, preventing any taxonomic changes or updates. Similarly, two other phylogenetic studies have included terminals of Ditaxinae, but these did not exceed 10% of taxon sampling and yielded low-resolution phylogenies [3,29].

Cervantes and collaborators reconstructed the biogeographic history of Acalyphoideae based on a molecular phylogenetic analysis using the *petD*, *trn*L-F and *mat*K/*trn*K genetic regions [30]. Ditaxinae, even though represented by ~10% of the species, emerged as paraphyletic. Their results recovered *Philyra* as a sister to the tribe Adelieae G.L.Webster, and this clade was a sister to the *Argythamnia* + *Chiropetalum* + *Ditaxis* clade*,* as shown by Jestrow [29,31]. *Caperonia* emerged as a sister to all above taxa, albeit with low support [30].

Given the need for a solid phylogenetic and systematic framework for the subtribe Ditaxinae, in this study we established the following aims: (1) test the monophyly of Ditaxinae and its currently recognized genera, *Argythamnia*, *Caperonia*, *Chiropetalum*, *Ditaxis* and *Philyra*, using a comprehensive taxonomic and geographical sampling, including multiple accessions per species when possible; (2) circumscribe the recovered clades and identify potential morphological synapomorphies; (3) establish a suprageneric classification in the subfamily Acalyphoideae based on the recovered phylogenetic pattern in this study.

## 2. Material and Methods

### 2.1. Taxon Sampling and Outgroup Selection

Our sampling covered all currently recognized genera of subtribe Ditaxinae: *Argythamnia* (11 spp., 61% of the total), *Caperonia* (10 spp., 30%), *Chiropetalum* (18 spp., 86%), *Ditaxis* (35 spp., 77%) and *Philyra* (1 sp., 100%). Thus, our dataset included a total of 75 species, representing 60% of Ditaxinae. We also included five representatives of tribe Adelieae, the latter based on Jestrow’s circumscription [29]. We used *Acalypha lanceolata* Willd., *Enriquebeltrania crenatifolia* (Miranda) Rzed., *Bernardia dichotoma* (Willd.) Müll.Arg., *Plukenetia penninervia* Müll.Arg., *P. volubilis* L. and *Seidelia triandra* (E.Mey.) Pax as outgroups, based on previous phylogenetic analyses [15,28,30]. Overall, our study sampled 86 species of the subfamily Acalyphoideae (Appendix A). The choice of outgroups also aimed to reconstruct the clades close to Ditaxinae following the study of Cervantes and collaborators [30]. The type species of each genus in Ditaxinae was sampled in our dataset. For the taxonomic treatment, type specimens were also analyzed to infer morphological similarities, mainly of taxa not represented in the phylogenetic analyses, in order to assess the preliminary generic assignment based on morphological similarities with the taxa represented in the phylogeny, as has been done in other complex groups within Euphorbiaceae [32,33,34].

We included samples collected in Africa, the Caribbean region, Central America, North America and South America. Plant tissues were preserved in silica gel, and vouchers were deposited in the herbaria BAA, FLOR, HUEFS, ICN, MA, MEXU, RB, SP, SPF and US (acronyms follow Thiers, continuously updated) [35]. Other tissue samples were obtained from herbarium specimens at BA, BAA, CA, CORD, CPAP, F, HUEFS, IEB, K, LPD, MA, MEXU, MO, MOL, RB, RSA, SI, SP, US and XAL (Appendix A). We also used 61 sequences (representing 22 species) from the US National Center for Biotechnology Information (NCBI) GenBank repository (https://www.ncbi.nlm.nih.gov/genbank). Voucher information and GenBank accession data are provided in Appendix A.

### 2.2. DNA Extraction, Amplification and Sequencing

DNA was extracted from silica-dried leaf tissue and herbarium material using the CTAB method [36] with some modifications [36] (see Appendix A). The extracted DNA was quantified using a Qubit™ dsDNA BR Standard (Invitrogen). Samples with high concentrations (>20 ng/μL) were diluted (1:20, 1:50) depending on the concentration.

Three plastid (*trn*L-F, *trn*T-L, *pet*D) and two nuclear (ITS, ETS) genetic regions were sequenced (see Appendix A). PCR amplifications were conducted with 25 μL reactions (for thermocycler temperature protocols, see Appendix A). Each reaction tube included MyTaq Red Mix (Bioline), H_2_O, primers and genomic DNA. For samples that were difficult to amplify, PuReTaq Ready-To-Go PCR Beads (GE Healthcare) were used. PCR products were purified with ExoSap PCR Purification and sent for sequencing at MACROGEN (Macrogen, Madrid, Spain), using the same amplification primers (Appendix A).

The pherograms were edited manually in UGENE [37] and automatically aligned with MUSCLE, using the default parameters. Manual adjustments were made to each alignment matrix in UGENE, employing the similarity criterion. A 120 bp region was excluded from the analysis of the *trn*T-L data matrix due to an uncertain homology assessment in the alignment.

### 2.3. Phylogenetic Analyses

Evolutionary models of nucleotide substitution were selected based on maximum likelihood (ML) using the Akaike (AIC) [38] information criterion implemented in jModelTest v.2.1.10 [39,40]. Each marker was analyzed individually, and the models were GTR + I + G for ETS and ITS, TVM + I + G for *trn*L-F, TPM1uf + G for *trn*T-L and GTR + G for *pet*D. MrBayes does not allow implementing all of these models, and thus, we used the nearest and slightly more complex model, which was GTR + I + G for the nuclear regions and GTR + G for the plastid markers [41]. Bayesian inference (BI) appears to be more robust with respect to over-parametrization and more sensitive to infra-parametrization than the ML optimization used in jModelTest [42]. Each genetic region was analyzed individually based on BI and ML. Concatenated matrices with nuclear (ITS + ETS) and plastid markers (*trn*L-F + *trn*T-L + *pet*D) were also analyzed separately to check for possible incongruences in the topology, and finally, a matrix with all markers was analyzed with BI and ML approaches. Topological incongruence between nuclear and plastid regions was defined as the presence of clades with a posterior probability (PP) ≥ 0.95 in IB and bootstrapping support (BS) ≥ 70% in ML [43]. In the combined analysis using only one terminal per species, we prioritized keeping the terminals with at least one nuclear and one plastid region. Bayesian analyses consisted of two independent Markov Chain Monte Carlo (MCMC) runs of 50 million generations in MrBayes v.3.1.2 [44], sampling every 1000th generation, with 20% (first 10 million trees) discarded as burn-in. Output files were summarized with TreeAnnotator v.1.6.1 [45], and the performance of each analysis (effective sample sizes, ESS > 200) was evaluated using Tracer v.1.6 [46]. Phylogenetic trees for individual and combined markers reconstructed with BI and ML are presented. Maximum-likelihood analyses were performed with RAxML [47] on the concatenated supermatrix, under a GTRGAMA model with 1000 bootstrap replicates. All analyses were hosted at CIPRES Science Gateway [48]. 

## 3. Results

The aligned DNA matrix combining the five regions (ETS, ITS, *pet*D, *trn*T-L, *trn*L-F) was 3985 bp long and included 86 species (75 of Ditaxinae *s.l.*) and 223 terminals (there were species represented by more than one specimen and unidentified/unnamed specimens labeled as “sp.”). A summary of each data partition and combined matrices is provided in Table 1. The marker *pet*D proved informative for the group. However, it was the region with the lowest taxonomic representation, as only recent tissue samples dried in silica gel could be amplified (Table 1). The analyses of the individual markers showed few cases of topological incongruences between the plastid and nuclear genome. However, in most cases, these incongruences did not have high support, and thus, the matrices (nuclear plus plastid datasets) were combined for the final analysis. Figure 1 represents the phylogenetic tree reconstructed when combining the five markers and the inclusion of one terminal per species. The phylogenetic analyses using all terminals (including multiple accessions) and individual and combined datasets are presented in Appendix A. The ML analysis did not show significant differences in tree topology when compared to the BI (Appendix A).

Despite minor incongruences between different reconstructions, the genera *Argythamnia*, *Caperonia*, *Chiropetalum*, *Philyra* and *Adelia* were confirmed as monophyletic, whereas *Ditaxis* was paraphyletic in all reconstructions (Figure 1 and Appendix A). In contrast, the phylogenetic trees obtained from the analyses of the combined and individual markers presented some incongruence regarding the positioning of *Philyra*, *Adelia* and *Caperonia*. In the analyses of the *trn*L-F and cpDNA-combined datasets, *Philyra* emerged as a sister (PP = 1) to the clade *Caperonia* + *Adelia* + *Chiropetalum* + *Argythamnia + Ditaxis* (Appendix A), whereas in *trn*T-L, *Philyra* formed a polytomy with *Adelia* (Appendix A). In *pet*D and ETS, *Adelia* + *Philyra* was a sister of *Caperonia* + *Chiropetalum* + *Argythamnia + Ditaxis* with maximum support (Appendix A). In the reconstructions based on ITS, ITS + ETS and the matrix with all markers combined, *Philyra* + *Adelia* emerged as a sister to *Argythamnia* + *Chiropetalum* + *Ditaxis*, while *Caperonia* emerged as a sister to the clade formed by all the five genera above (Figure 1 and Appendix A). Based on ETS only, *Caperonia* emerged as a sister to *Argythamnia* + *Ditaxis*, while *Chiropetalum* was recovered as a sister to *Caperonia* + *Argythamnia* + *Ditaxis*, both with low support (Appendix A). In all other analyses, *Chiropetalum* emerged as a sister to *Argythamnia* + *Ditaxis* with high support (Figure 1 and Appendix A). In all reconstructions, *Ditaxis* species were grouped into two clades (*Ditaxis* 1 and *Ditaxis* 2) separated by *Argythamnia s.s.* (Figure 1), leaving *Ditaxis* paraphyletic in its current circumscription. In ETS and ITS + ETS reconstructions (Appendix A), the Andean species *Ditaxis jablonszkyana* Pax & K.Hoffm. and *D. malpighipila* (Hicken) L.C.Wheeler emerged as sisters to all other *Ditaxis* + *Argythamnia* species (PP = 1), whereas in all plastid reconstructions, these two species were recovered as sisters to clade *Ditaxis* 2 (PP = 1, Figure 1; Appendix A). 

Phylogenetic trees generated from nuclear and plastid datasets, based on both BI and ML, supported the paraphyly of Ditaxinae as currently circumscribed (Figure 1 and Appendix A) due to the position of the representatives of the Adelieae tribe between the terminals of Ditaxinae. The results also reinforce that the Chrozophoreae tribe is polyphyletic in the current circumscription (Figure 1).

All species of *Chiropetalum* formed a single clade, and the geographically disjunct Mexican species emerged together with South American species. The largest clade of Ditaxinae (*Argythamnia* + *Ditaxis*) was recovered as the sister of *Chiropetalum*. *Argythamnia* species, all from the central region of the Americas (Caribbean, Central America and southern Mexico), resulted as the sister clade of *Ditaxis* species (*Ditaxis* 1 clade) with North American distribution (Figure 1). The clade *Ditaxis* 2, the sister of *Ditaxis* 1 + *Argythamnia*, included North American and all Central and South American species. The five African species/specimens of *Caperonia* sampled in the phylogeny (identified with * in Figure 1) were placed in two different clades (Figure 1). Species of the tribe Adelieae, exclusive to Central and South America, emerged as the sister clade of the monospecific genus *Philyra* (Figure 1).

## 4. Discussion

This study presents the most comprehensive taxonomic and geographical sampling of Ditaxinae (ca. 60%) to date. In an attempt to solve the generic relationships among *Argythamnia*, *Chiropetalum* and *Ditaxis*, Ramírez-Amezcua [28] and Külkamp [15] sampled approximately 30% and 25% of the Ditaxinae species, respectively. Furthermore, the sampling of related groups (*Caperonia* and Adelieae) was less than 5%, precluding any suprageneric taxonomic decisions. As a result, our research provides a solid phylogenetic framework for new taxonomic delimitations at the genus and tribe levels.

### 4.1. Changes in Generic Delimitation

The relationship between *Argythamnia* and *Ditaxis* could not be resolved in previous studies, probably because of the relatively low (30%) taxon sampling and lack of phylogenetic support for some clades [15,28], while the genus *Chiropetalum*, albeit with low support, emerged as a separated clade in both studies. A recent phylogenetic reconstruction using a large representation of subfamily Acalyphoideae [30] also recovered a monophyletic *Chiropetalum*, in this case with maximum support. Here, in all reconstructions, *Chiropetalum* emerged as monophyletic and a sister to the clade containing *Ditaxis* and *Argythamnia*, with maximum support (PP = 1) (Figure 1). The high taxon sampling of *Chiropetalum* (90%) in our phylogenetic analyses gives us confidence in circumscribing the genus as a distinct taxon. However, further phylogenetic studies should sample *Chiropetalum patagonicum* (Speg.) O’Donell & Lourteig, since the species presents a remarkable divergent morphology (prostrate habit, absence of trichomes, petals of the staminate flower slightly lobed) from that of the rest of *Chiropetalum*. Ingram treated this species in the genus *Aonikena* Apeg. [23], whereas O’Donell & Lourteig classified it in *Chiropetalum* sect. *Aonikena* (Speg.) O’Donell & Lourt [49]. *Aonikena patagonica* would be well placed in *Chiropetalum* based on comparative morphology, but nevertheless, the inclusion of this species in further phylogenetic analyses is still required to definitively clarify its taxonomic placement. Based on our phylogenetic reconstruction and morphology studies, we identified several synapomorphies of *Chiropetalum*, including lobed petals in the staminate flowers (Figure 2C), stamens disposed in a whorl and fused at the base forming a column (Figure 2C) and the absence of petals in the pistillate flowers (Figure 2D), except for *C. tricuspidatum* (Lam.) A.Juss. and *C. argentinense* Skottsb., which have vestigial petals [23,26,49]. A few species of *Argythamnia s.s.* also have pistillate flowers without petals [22]. The presence of stellate trichomes is also a unique feature of *Chiropetalum*, but these trichomes are present in only 10 species (50%) [23,26].

Our results show that *Ditaxis* as currently recognized is paraphyletic, because the species of *Argythamnia s.s.* are nested within *Ditaxis* (Figure 1), a topology similar to the phylogenetic reconstruction in Ramirez-Amezcua [28]. The staminate flowers in *Argythamnia* have four (rarely five) petals and four (rarely five) free stamens, whereas in *Ditaxis* the staminate flowers present with five petals and 8–10 stamens united in a column. Thus, to avoid describing a new genus lacking morphological synapomorphies or a clear set of distinguishing characteristics, we expanded the circumscription of *Argythamnia s.s.* with the inclusion of the two clades of *Ditaxis* (clade 1 & 2; Figure 1) following, in part, Ingram’s classification system [27]. Thus, *Argythamnia* in the circumscription proposed here is monophyletic and composed of three well-supported clades (Figure 1): (i) *Argythamnia s.s.*, (ii) *Ditaxis* clade 1, exclusive to North America, and (iii) *Ditaxis* clade 2, the most diverse clade of *Ditaxis s.s*., with a distribution from North America to southern South America. In this new classification framework, *Argythamnia s.l.* is supported by the presence of petals in pistillate flowers (Figure 2A) (rarely absent) and entire petals (unlobed) in staminate flowers (Figure 2B). The presence of an apiculum on the seeds of *Argythamnia s.l.* should be studied further. Due to the lack of specimens with seeds for nine species of *Argythamnia s.l.*, that structure was little explored in this study. The seeds of the other genera of the tribes Ditaxeae, Adelieae and Caperonieae are globose rather than apiculate.

*Caperonia sensu* Webster [5] is the only genus of Ditaxinae with an extra-New World distribution, with seven species occurring in tropical Africa and Madagascar. Here, we confirmed *Caperonia* as monophyletic, as suggested by Cervantes and collaborators [30], but with a broader taxon sampling. Pax & Hoffmann proposed two sections for *Caperonia*, *C.* sect. *Eucaperonia* ([nom. invalid.], autonym section = sect. *Caperonia*) and *C.* sect. *Aculeolatae* Pax & K.Hoffm. (taxa with prickles sampled in our phylogeny, *C. corchoroides* Müll.Arg., *C. cordata* A.St.-Hil., *C. heteropetala* Didr., *C. linearifolia* A.St.-Hil.) [14]. Based on morphology, we would have expected that these sections to be recovered in two clades in our phylogenetic analyses, but the presence of prickles appears to represent a plesiomorphic state, and some of the taxa studied have lost this state independently. However, we emphasize that *Caperonia* requires additional research with a larger taxonomic representation to clarify phylogenetic relationships, explore the need to establish an infrageneric classification and understand the origin and nature (multiple or single colonization events) of its amphi-Atlantic distribution pattern. When comparing *Caperonia* with phylogenetically closely related genera, its morphological divergence is marked by the presence of glandular trichomes (Figure 2H), a muricate ovary surface (Figure 2G) and parallel secondary veins (Figure 2I). These features are absent in all the other genera and are recognized here as synapomorphies for *Caperonia*. Another contrasting characteristic of *Caperonia* is its exclusive occurrence in marshy habitats [17,21], while all other related genera are found in desert or seasonally dry environments [15,16,19,22,23,24,25].

*Philyra brasiliensis* was originally the only species described in *Philyra*; however, the species was combined in *Ditaxis* by Baillon [50] and later transferred to *Argythamnia* by Müller [13]. Morphology does not support these classifications because *Philyra* lacks the synapomorphies recognized for *Argythamnia* + *Chiropetalum* (presence of floral nectaries and malpighiaceous trichomes). Moreover*, Philyra* is the only genus of the focal taxa having a pair of spines inserted on branches beneath the leaves (Figure 2F). Because of these unique characteristics, the species was treated again in *Philyra* [26]. The phylogenetic analyses of Jestrow and collaborators [31], Cervantes and collaborators [30] and our own results also support the circumscription of *Philyra* as a monospecific genus. The genus *Adelia* (sister to *Philyra*) includes some species with pointed branches, but it lacks the pair of spines below the leaves. *Adelia* is also distinguished from *Philyra* by its apetalous staminate flowers clustered in glomerules (Figure 2E), whereas in *Philyra*, the staminate flowers are dichlamydeous and grouped in racemes and the stamens (10–12) form a column with two whorls. Detailed phylogenetic information about *Adelia* can be found in previous studies focused on Adelieae that included a larger taxonomic representation [29,31,51,52].

### 4.2. Tribe Delimitation

Before our study, taxonomic affinities and phylogenetic relationships of subtribe Ditaxinae were uncertain mainly due to the poor taxon sampling in previous phylogenetic analyses [15,28,30,31,52]. Our results showed a robust topology (Figure 1), allowing us to propose a new classification. Ditaxinae has traditionally been assigned to the Chrozophoreae tribe [5,7,10]. Other phylogenetic analyses, however, revealed Chrozophoreae to be polyphyletic and Ditaxinae to be paraphyletic [2,30,31,51]. Here, we confirmed both results, with tribe Adelieae recovered as embedded among the terminals of Ditaxinae (Figure 1). *Argythamnia* (including *Ditaxis*) and *Chiropetalum* are part of Ditaxinae, which appear to be more closely related to each other than to *Caperonia* and *Philyra* (Figure 1 and Figure 2).

Following our phylogenetic framework, we elevated Ditaxinae to the rank of tribe (Ditaxeae), including the genera *Argythamnia* (including *Ditaxis*) and *Chiropetalum* (Figure 1 and Figure 2) and excluding *Caperonia* and *Philyra* (see the taxonomic treatment below). Tribe Ditaxeae is supported by two synapomorphies: the presence of floral nectaries (Figure 2C) and malpighiaceous trichomes (Figure 2J). Another important characteristic is the presence of a basal and suprabasal actinodromous venation pattern, which is very similar among taxa, but with small variations regarding the number of basal secondary veins (2–4) and the intensity of their impression on the leaf’s surface. However, this character is not exclusive to Ditaxeae; some taxa in the tribe Adelieae also present a similar venation pattern. With the exclusion of subtribe Ditaxinae, the tribe Chrozophoreae is now circumscribed to include subtribes Speranskiinae and Chrozophorinae, which are exclusively paleotropical in their distribution.

We propose to circumscribe *Philyra* within tribe Adelieae (Figure 1 and Figure 2), as suggested by Jestrow [31,51]. Traditionally, *Philyra* was circumscribed in Chrozophoreae and not in Adelieae, supported by the presence of petals in the pistillate and staminate flowers [5,10]. Now, tribe Adelieae comprises the genera *Adelia*, *Garciadelia* Jestrow & Jiménez Rodr., *Lasiocroton* Griseb., *Leucocroton* Griseb. and *Philyra*, which are united by two synapomorphies: the dioecious sexual system (rarely monoecious in *Leucocroton*) and the arborescent to shrubby habit (Figure 2K,L). 

Systematists have always had difficulty placing *Caperonia*. Klotzsch [53] classified *Caperonia* in tribe Crotoneae Dumort., whereas Müller [13] placed it within tribe Acalypheae Dumort., subtribe Caperoniinae Müll.Arg. Pax & Hoffmann [14], including the genus in subtribe Chrozophorinae, and Webster [7] classified *Caperonia* as part of tribe Chrozophoreae, subtribe Ditaxinae, where it remained until now. Here, we circumscribe *Caperonia* as a monogeneric tribe based on strong phylogenetic and morphological evidence. In the most recent phylogenetic reconstruction, based on plastid data only, *Caperonia* emerged as a sister to *Argythamnia* + *Chiropetalum* + *Ditaxis* + Adelieae [30]. Although we found that the position of *Caperonia* was incongruent (but with low support) among phylogenetic reconstructions based on individual plastid and nuclear markers (Appendix A), our combined analysis provides strong support for its position as a sister to Adelieae + Ditaxineae (as circumscribed here), justifying its treatment as a monogeneric tribe, Caperonieae.

The new tribe Caperonieae (see taxonomic treatment below) is supported by the presence of glandular trichomes (Figure 2H) and a muricate ovary surface (Figure 2G). We also highlight the presence of leaves with craspedodromous secondary veins (Figure 2I), heteromorphic petals in staminate flowers in most species and a thickened structure at the apex of the staminal column, identified by some authors as a rudimentary ovary (pistillode) [5]. However, ontogenetic studies are needed to understand the origin of this floral structure.

### 4.3. Taxonomic Treatment

The molecular phylogenetic results presented here support the establishment of a new classification for Ditaxinae, raising it from the subtribe to the tribe level (Ditaxeae), and including two well-supported clades composed of genera *Chiropetalum* and *Argythamnia*. We maintain tribe Adelieae, extending its circumscription to include the genus *Philyra*. We also elevate subtribe Caperoniinae to the tribe level, adding two new tribes to the subfamily Acalyphoideae. Furthermore, we expanded the circumscription of *Argythamnia* to include the two well-supported clades of *Ditaxis*, representatives that emerged as paraphyletic in our analyses. Future studies will be directed at refining this delimitation and possibly proposing infrageneric classification systems for *Argythamnia*, *Caperonia* and *Chiropetalum*. Here, we present the names and diagnosis of the tribes and genera recognized, as well as a list of all species recognized under each genus. The necessary infrageneric nomenclature combinations will be presented in future taxonomic studies. Species with phylogenetic data used in this study are marked with an asterisk (*) in the “species recognized” section of each genus below. In Appendix A, we present a summary of the new and previous classification of all taxa treated here.


**1. CAPERONIEAE Külkamp & Riina, *stat. nov.***


Basionym: Caperoniinae Müll.Arg. (as ‘Caperonieae’), Linnaea 34: 152. 1865.

Type, designated here: *Caperonia* A.St.-Hil.

***Caperonia*** A.St.-Hil. Histoire des plantes les plus remarquables du Bresil et fu Paraguay 3/4: 244–247. 1825. *Ditaxis* sect. *Caperonia* (A.St.-Hil.) Baill. Adansonia, 4: 272. 1865.

**Description:** Monoecious, rarely dioecious; herbs, rarely subshrubs, annual or perennial; stems hollow; trichomes simple and glandular, sometimes prickly; stipules present; leaves alternate, petiolate or subsessile, penninerved, rarely palmatinerved, with craspedodromous secondary veins, margins serrate; inflorescences racemiform, bisexual or unisexual, bracteoles uniflorous, flowers dichlamydeous; staminate flowers with articulated pedicels; sepals 5, lanceolate, margin entire, pubescent or glabrous; petals 5, often unequal, glabrous, rarely pubescent, basally adnate to the staminal column; stamens 8–10 in two whorls, and pistillode on the column apex; floral nectaries absent; pistillate flowers proximal, dichlamydeous, sepals 5–6, equal or unequal, lanceolate to ovate, margin entire, pubescent, persistent in fruit; petals 5, usually equal, unequal or reduced; floral nectaries absent; ovary 3–locular, surface muricate, covered by glandular trichomes; style multifid; capsule verrucose, columella persistent; seeds one per locule, orbicular, foveolate, gray to black.

**Distribution:** *Caperonia* is distributed in the New World and Africa (continental Africa and Madagascar). The greatest diversity of *Caperonia* occurs in South America, mainly Brazil, with approximately 40% of the taxa (14). All *Caperonia* species occur in marshy environments [5,17,21].

**Species recognized** (35). **Africa/Madagascar** (7): *Caperonia fistulosa* Beille*, *C. latifolia* Pax, *C. palustris* (L.) A.St.-Hil.*, *C. rutenbergii* Müll.Arg., *C. serrata* (Turcz.) C.Presl.*, *C. stuhlmannii* Pax*, *C. subrotunda* Chiov. **America** (29): *Caperonia aculeolata* Müll.Arg., *C. altissima* Eskuche, *C. amarumayu* Külkamp & Cordeiro, *C. angustissima* Klotzsch, *C. bahiensis* Müll.Arg.*, *C. buettneriacea* Müll.Arg., *C. capiibariensis* Eskuche, *C. castaneifolia* (L.) A.St.-Hil.*, *C. castrobarrosiana* Paula & Hamburgo, *C. chiltepecensis* Croizat*, *C. corchoroides* Müll.Arg.*, *C. cordata* A.St.-Hil.*, *C. cubana* Pax & K.Hoffm.*, *C. gardneri* Müll.Arg., *C. glabrata* Pax & K.Hoffm., *C. heteropetala* Didr.*, *C. hystrix* Pax & K.Hoffm., *C. langsdorffii* Müll.Arg., *C. linearifolia* A.St.-Hil.*, *C. lutea* Pax & K.Hoffm., *C. maracaibensis* Külkamp & Cordeiro, *C. multicostata* Müll.Arg., *C. neglecta* G.L.Webster, *C. palustris* (L.) A.St.-Hil.*, *C. paraguayensis* Pax & K.Hoffm., *C. regnellii* Müll.Arg., *C. similis* Pax & K.Hoffm., *C. stenophylla* Müll.Arg. and *C. zaponzeta* Mansf.


**2. DITAXEAE Külkamp & Riina, *stat. nov.***


Basionym: Ditaxinae Griseb., Fl. Brit. W. I., 43. 1859.

Type, designated here: *Ditaxis* Vahl ex A.Juss.

**Description:** Monoecious rarely dioecious; herbs, annual or perennial, and shrubs; branches erect, decumbent or prostrate; stipules present; leaves simple, alternate; venation actinodromous basal and suprabasal; margins serrate to entire; trichomes malpighiaceous, simple or stellate in both surfaces; racemes axillary, bisexual, rarely unisexual; pistillate flowers proximal and staminate distal; bracteoles uniflorous, lanceolate to ovate, pubescent, rarely glabrous; staminate flowers dichlamydeous; sepals (4–)5, linear to lanceolate, margin entire or serrated, pubescent or glabrous; petals (4–)5, entire, erose, laciniate or lobed, glabrous or pubescent, adnate to the staminal column; stamens 4–10, distinct or connate forming a column, stamens in one or two whorls; staminodes 0–5 at the top of the staminal column, pubescent or glabrous; floral nectaries 4–5, pubescent or glabrous; pistillate flowers dichlamydeous or monochlamydeous; sepals (4–)5(–6), linear, lanceolate, ovate or elliptic, pubescent rarely glabrous; petals 0–5, linear, lanceolate, oval, elliptic or rhomboid, pubescent or glabrous, margins entire, erose or laciniate; floral nectaries 5, adnate to the receptacle at the base of the ovary, glabrous, ciliate or pubescent; ovary pubescent, rarely glabrous; styles bifid or trifid, pubescent or glabrous; capsule 3–locular, smooth, pubescent rarely glabrous; seeds one per locule, orbicular to ovoid, apiculate or not, surface foveolate, smooth, undulate or reticulate, gray to black.

**Distribution:** Species of Ditaxeae are distributed throughout the New World, from the southern United States to Patagonia in the south of Argentina. There are two main centers of diversity for Ditaxeae: the first comprising southern North America, the Caribbean Islands and northern and central South America, and the second in northeastern Brazil [15,16,18,20,22,23,24,25,28].

***Argythamnia*** P.Browne, Civ. Nat. Hist. Jamaica: 338. 1756.—Type: *Argythamnia candicans* Sw. = *Ditaxis* Vahl ex A.Juss., Euphorb. Gen. 27. 1824.—Type: *Ditaxis fasciculata* Vahl ex A.Juss.

**Description:** Monoecious, rarely dioecious herbs to shrubs, annual or perennial; trichomes malpighiaceous and simple; racemes bisexual, rarely unisexual; staminate flowers 2–15, dichlamydeous, sepals (4)5; petals (4)5, glabrous or pubescent, entire, erose or laciniate; stamens 4–10, distinct or connate, when connate arranged in two whorls; staminodes 0–5 at the top of the staminal column, pubescent or glabrous; floral nectaries 4 or 5, glabrous; pistillate flowers 1–4; dichlamydeous or monochlamydeous, sepals (4–)5(–6); petals 5, rarely 0, entire, erose or laciniate; floral nectaries 5, glabrous or ciliate; styles bifid or trifid; seeds orbicular to ovoid, apiculate, surface smooth, undulate or reticulate.

**Distribution:** *Argythamnia* is distributed throughout the New World, from southern United States to Patagonia. Greater diversity is found in southern North America, the Caribbean Islands, northern and central South America and northeastern Brazil [16,18,22,24,25,26,28].


**New combination**


*Argythamnia grazielae* (Külkamp) Külkamp & Riina **comb. nov.**

≡ *Ditaxis grazielae* Külkamp, Phytotaxa 455(1): 154. 2020.

Type: BRAZIL. Bahia: Wanderley, 25 January 1996 (fl. fr.), *B.R. Chagas s.n.* (holotype: RB [RB00084882]!; isotypes: CEPEC [CEPEC131190]!, K [K001206888]!, MG!, NY [NY01183998]!, SPF [SPF196837]!).

**Species recognized** (68): *Argythamnia acaulis* (Herter ex Arechav.) J.W.Ingram, *A. acutangula* Croizat*, *A. adenophora* A.Gray*, *A. aphoroides* Müll.Arg.*, *A. argentea* Millsp., *A. argothamnoides* (Bertero ex Spreng.) J.W.Ingram*, *A. argyraea* Cory*, *A. arlynniana* J.W.Ingram, *A. brandegeei* Millsp.*, *A. breviramea* Müll.Arg.*, *A. calycina* Müll.Arg., *A. candicans* Swartz*, *A. claryana* Jeps.*, *A. coatepensis* (T.S.Brandegee) Croizat*, *A. cubensis* Britton & Wilson*, *A. cuneifolia* (Pax & K.Hoffm.) J.W.Ingram, *A. cyanophylla* (Wooton & Standl.) J.W.Ingram*, *A. depressa* (Greenm.) J.W.Ingram, *A. desertorum* Müll.Arg.*, *A. dioica* (Bonpland, Humboldt & Kunth) Müll.Arg.*, *A. ecdyomena* J.Ingram*, *A. erubescens* J.R.Johnst., *A. fasciculata* (Vahl ex A.Juss.) Müll.Arg.*, *A. fendleri* Müll.Arg., *A. grazielae* (Külkamp) Külkamp & Riina*, *A. guatemalensis* Müll.Arg.*, *A. haitiensis* (Urb.) J.W.Ingram, *A. haplostigma* Pax & K.Hoffm., *A. heterantha* (Zucc.) Müll.Arg.*, *A. heteropilosa* J.W.Ingram, *A. humilis* (Engelm. & A.Gray) Müll.Arg.*, *A. illimaniensis* (Baill.) Müll.Arg., *A. ingramii* Y.Ramírez-Amezcua & V.W.Steinm., *A. jablonszkyana* (Pax & K.Hoffm.) J.W.Ingram*, *A. katharinae* (Pax) Croizat, *A. lanceolata* (Benth.) Müll.Arg.*, *A. lottiae* J.W.Ingram*, *A. lucayana* Millsp.*, *A. lundellii* J.W.Ingram*, *A. macrantha* (Pax & K.Hoffm.) Croizat, *A. macrobotrys* (Pax & K.Hoffm.) J.W.Ingram, *A. malpighiacea* Ule*, *A. malpighiphila* (Hicken) J.W.Ingram*, *A. manzanilloana* Rose*, *A. mercurialina* (Nutt.) Müll.Arg.*, *A. microphylla* Pax, *A. montevidensis* (Didr.) Müll.Arg.*, *A. moorei* J.W.Ingram*, *A. oblongifolia* Urb., *A. pilosissima* (Benth.) Müll.Arg., *A. polygama* (Jacq.) Kuntze*, *A. pringlei* Greenm.*, *A. proctorii* J.W.Ingram, *A. purpurascens* S.Moore*, *A. rubricaulis* (Pax & K.Hoffm.) Croizat*, *A. salina* (Pax & K.Hoffm.) J.W.Ingram*, *A. sellowiana* (Pax & K.Hoffm.) J.W.Ingram*, *A. sericea* Griseb.*, *A. serrata* (Torr.) Müll.Arg.*, *A. silviae* Y.Ramírez-Amezcua & V.W.Steinm.*, *A. simoniana* (Casar.) Müll.Arg.*, *A. simulans* J.W.Ingram*, *A. sitiens* (T.S.Brandegee) J.W.Ingram, *A. stahlii* Urb., *A. tinctoria* Mill.* and *A. wheeleri* J.W.Ingram*.

***Chiropetalum*** A.Juss., Ann. Sci. Nat. (Paris) 25: 21. 1832. —Type: *Chiropetalum lanceolatum* (Cav.) A.Juss.

**Description:** Monoecious herbs or subshrubs, perennial rarely annual; trichomes malpighiaceous, simple, stellate or rarely absent (*C. patagonicum*); racemes bisexual, rarely unisexual; staminate flowers 3–35, dichlamydeous, sepals 5; petals 5, glabrous, 3–7-lobed; floral nectaries 5, glabrous or pubescent; stamens 3–6, partially connate forming a column, anthers arranged in one whorl, staminodes absent; pistillate flowers 1–5, monochlamydeous, rarely dichlamydeous, sepals 5; petals usually absent, rarely 5; floral nectaries 5, glabrous or pubescent; styles bifid; capsule covered by simple, stellate and/or malpighiaceous trichomes; seeds orbicular, surface foveolate or rough.

**Distribution:** *Chiropetalum* is distributed in South America (19 species) and Mexico (2 species). Species richness is concentrated in the central region of South America, and the species presenting with the southernmost distribution is *C. patagonicum*, occurring in the Patagonia region of Argentina. Morphological and geographic details of each species can be found in studies of Ingram and Külkamp [15,20,23,26].

**Species recognized** (21): *Chiropetalum anisotrichum* (Müll.Arg.) Pax & K.Hoffm.*, *C. argentinense* Skottsb.*, *C. astroplethos* (J.W.Ingram) Radcl.-Sm. & Govaerts*, *C. berteroanum* Schltdl.*, *C. boliviense* (Müll.Arg.) Pax & K.Hoffm.*, *C. canescens* Phil.*, *C. cremnophilum* I.M.Johnst., *C. foliosum* Pax & K.Hoffm.*, *C. griseum* Griseb.*, *C. intermedium* Pax & K.Hoffm.*, *C. molle* (Klotzsch ex. Baill.) Pax & K.Hoffm.*, *C. patagonicum* (Speg.) O’Donell & Lourteig, *C. pavonianum* (Müll.Arg.) Pax, *C. phalacradenium* (J.W.Ingram) L.B.Sm. & Downs*, *C. puntaloberense* Alonso Paz & Bassagoda*, *C. quinquecuspidatum* (A.Juss.) Pax & K.Hoffm.*, *C. ramboi* (Allem & Irgang) Radcl.-Sm. & Govaerts*, *C. ruizianum* (Müll.Arg.) Pax & K.Hoffm., *C. schiedeanum* (Müll.Arg.) Pax*, *C. tricoccum* (Vell.) Chodat & Hassl.* and *C. tricuspidatum* (Lam.) A.Juss*.


**3. ADELIEAE G.L.Webster Taxon 24: 597. 1975**


Type: *Adelia* L.

**Description:** Dioecious, rarely monoecious; trees to shrubs; pair of stipular spines absent, rarely present (*Philyra*); leaves alternate, simple; penninerved or actinodromous, basal and suprabasal, margins dentate to entire; trichomes simple or stellate; inflorescences axillary in racemes, glomerules or subpanicles, unisexual, rarely bisexual; staminate flowers monochlamydeous or dichlamydeous; sepals 4–5; petals absent, rarely 5; entire, pubescent; stamens 8–18(–30), filaments distinct or connate at base; staminode present or absent, floral nectaries absent; pistillate flowers monochlamydeous or dichlamydeous, sepals 5–6 lanceolate, ovate or elliptic, pubescent, rarely glabrous, persistent in fruit; petals 0–5, lanceolate, oval, elliptic or rhomboid, pubescent or glabrous; floral nectaries absent; ovary 3-locular, pubescent; ovules with inner integuments thick, outer thin to thick; styles bifid to multifid, pubescent or glabrous; capsule 3–locular; columella persistent; seeds one per locule, orbicular, surface foveolate, rough or smooth.

**Distribution:** Adelieae taxa are found from Mexico to Argentina, with three of the five genera endemic to the West Indies (*Garciadelia*, *Lasiocroton* and *Leucocroton*) [19,26,29,31,51,52,54,55].

***Philyra*** Klotzsch, Archiv für Naturgeschichte 7(1): 199. 1841.—Type: *Philyra brasiliensis* Klotzsch

**Description:** Dioecious shrubs or treelets; paired stipular spines; leaves glabrous, venation pinnate, margin entire; bracts paleaceous, pubescent to glabrescent; staminate flowers dichlamydeous; sepals 5, petals 5; stamens 10–12, connate in a column with 2 whorls; staminodes 2 at the top of column, pubescent; pistillate flowers with pedicels larger than 12 mm long, dichlamydeous, petals 5, larger than sepals, styles multifid; capsule glabrous, columella persistent; seeds orbicular, surface smooth, gray to blackish.

**Distribution:** *Philyra* is distributed in northern Argentina, central and southern Paraguay and Brazil. In Brazil, this species occurs in the central–western region and the Atlantic coast, in the southeast and northeast of the country. For additional information, see Külkamp’s studies [19,26].

**Species recognized** (1): *Philyra brasiliensis* Klotzsch*.

***Adelia*** L., Syst. Nat. ed. 10, 2: 1298. 1759 (nom. cons.).—Type: *Adelia ricinella* L. (*typ. cons.*)

**Distribution:** *Adelia* has a continuous distribution from the southern United States to central South America. The greatest diversity of species is found in Mexico and Central America. Although a few species are widespread (e.g., *Adelia membranifolia* (Müll.Arg.) Chodat & Hassl.), most of them have a narrow distribution, and some are only known to be from a limited number of localities (e.g., *Adelia cinerea* (Wiggins & Rollins) A.Cerv., V.W.Steinm. & Flores-Olivera).

For a diagnosis, see De-Nova & Sosa and Jestrow’s studies [31,51,55].

**Species recognized** (9): *Adelia barbinervis* Cham. & Schltdl., *A. brandegeei* V.W. Steinm.*, *A. cinerea* (Wiggins & Rollins) A.Cerv., V.W.Steinm. & Flores-Olvera*, *A. membranifolia* (Müll.Arg.) Chodat & Hassl., *A. oaxacana* (Müll.Arg.) Hemsl.*, *A. obovata* Wiggins & Rollins, *A. ricinella* L., *A. triloba* (Müll.Arg.) Hemsl.* and *A. vaseyi* (J.M.Coult.) Pax & K.Hoffm.

***Garciadelia*** Jestrow & Jiménez Rodr., Taxon 59(6): 1809–1810. 2010.—Type: *Croton leprosus* Willd.

**Distribution:** *Garciadelia* has four species endemic to Hispaniola.

For a diagnosis, see Jestrow’s studies [29,31].

**Species included** (4): *Garciadelia abbottii* Jestrow & Jiménez Rodr., *G. castilloae* Jestrow & Jiménez Rodr., *G. leprosa* (Willd.) Jestrow & Jiménez Rodr. and *G. mejiae* Jestrow & Jiménez Rodr.

***Lasiocroton*** Griseb., Fl. Brit. W. I. 46. 1864.—Type: *Croton macrophyllus* Sw.

**Distribution:** *Lasiocroton* occurs in Cuba, Hispaniola, Jamaica and the Bahamas.

For a diagnosis, see Jestrow’s studies [29,31].

**Species recognized** (7): *Lasiocroton bahamensis* Pax & K.Hoffm., *L. fawcettii* Urb., *L. gracilis* Britton & P.Wilson, *L. gutierrezii* Jestrow, *L. harrisii* Britton, *L. macrophyllus* (Sw.) Griseb. and *L. microphyllus* (A.Rich.) Jestrow.

***Leucocroton*** Griseb. Abh. Königl. Ges. Wiss. Göttingen, 9: 20. 1861.—Type: *Leucocroton wrightii* Griseb.

**Distribution:** *Leucocroton* is restricted to serpentine soils of Cuba.

For a diagnosis, see studies by Borhidi and Jestrow [29,31,54].

**Species recognized** (26): *Leucocroton acunae* Borhidi, *L. anomalus* Borhidi, *L. bracteosus* Urb., *L. brittonii* Alain, *L. comosus* Urb., *L. cordifolius* (Britton & P.Wilson) Alain, *L. discolor* Urb., *L. ekmanii* Urb., *L. flavicans* Müll.Arg., *L. havanensis* Borhidi, *L. incrustatus* Borhidi, *L. linearifolius* Britton, *L. longibracteatus* Borhidi, *L. moaensis* Borhidi & O.Muñiz, *L. moncadae* Borhidi, *L. obovatus* Urb., *L. pachyphylloides* Borhidi, *L. pachyphyllus* Urb., *L. pallidus* Britton & P.Wilson, *L. revolutus* C.Wright, *L. sameki* Borhidi, *L. saxicola* Britton, *L. stenophyllus* Urb., *L. subpeltatus* (Urb.) Alain, *L. virens* Griseb. and *L. wrightii* Griseb.

## Figures and Tables

**Figure 1 biology-12-00173-f001:**
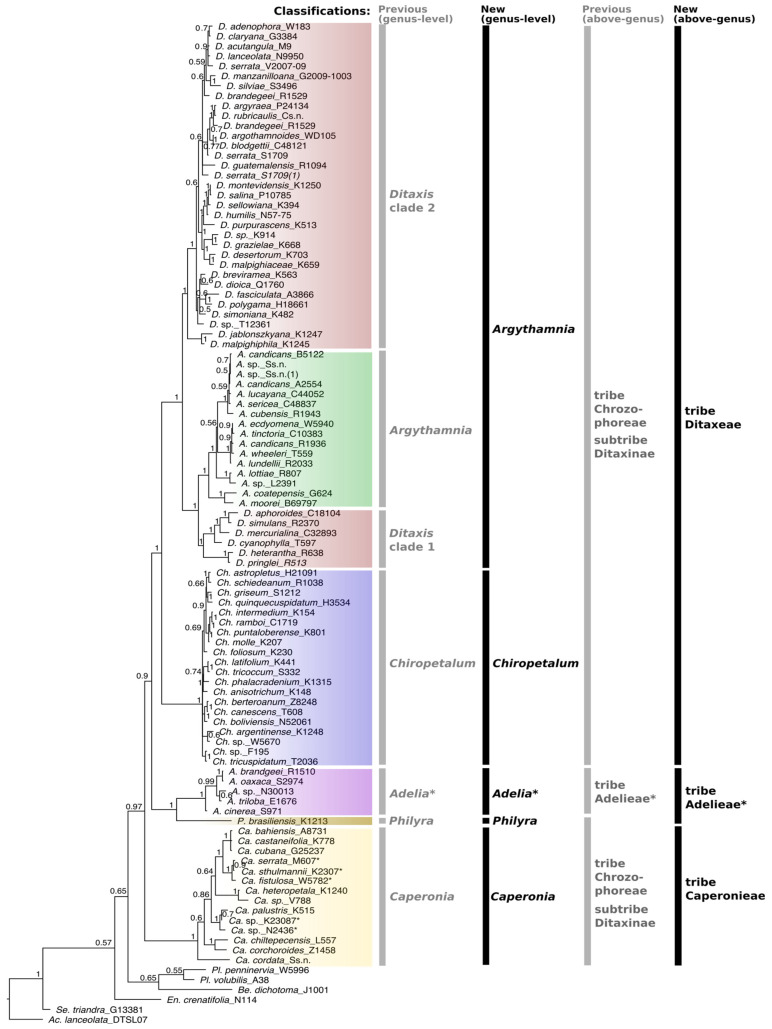
Majority rule consensus tree of Ditaxinae (Euphorbiaceae) and related taxa based on the combined five markers (cpDNA [*trn*LF, *trn*TL, *pet*D] and nDNA [ETS, ITS]) obtained through Bayesian inference. Bayesian posterior probabilities (PPs) are indicated on each branch. Vertical bars with labels on the right indicate the old (gray) [5] and the new (black) generic and suprageneric classifications. Asterisks in *Adelia* and Adelieae indicate that this clade is not fully represented here (several unsampled genera); we followed the classification proposed by Jestrow [29] (based on a complete generic sampling of Adelieae). To avoid confusion, some genera are abbreviated using two initial letters: Ac. = *Acalypha*, Be. = *Bernardia*, Ca. = *Caperonia*, Ch. = *Chiropetalum*, En. = *Enriquebeltrania*, Pl. = *Plukeneria*, Se. = *Seidelia*.

**Figure 2 biology-12-00173-f002:**
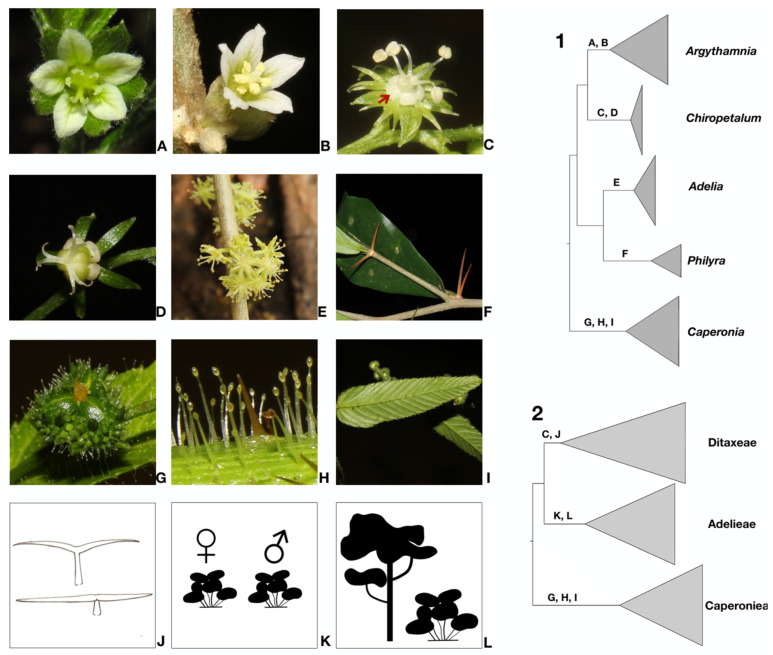
Schematic phylogenies of Ditaxinae and related taxa based on Bayesian inference using the combined five markers (cpDNA [*trn*LF, *trn*TL, *pet*D] and nDNA [ETS, ITS]). **1** & **2.** Generic (1) and suprageneric (2) classifications proposed here. Letters on branches indicate the morphological synapomorphies supporting each clade corresponding to the following illustrations. (**A**) Dichlamydeous pistillate flower of *Argythamnia desertorum*. (**B**) Staminate flower with entire petals of *Argythamnia desertorum*. (**C**) Staminate flower with lobed petals of *Chiropetalum phalacradenium* for cladogram 1 and floral nectaries for cladogram 2. (**D**) Monochlamydeous pistillate flower of *Chiropetalum phalacradenium*. (**E**) Monochlamydeous staminate flowers of *Adelia membranifolia*. (**F**) Pair of thorns below the leaves in *Philyra brasiliensis*. (**G**) Ovary with muricate surface in *Caperonia heteropetala*. (**H**) Glandular trichomes in *Caperonia heteropetala*. (**I**) Leaves with craspedodromous secondary veins in *Caperonia heteropetala*. (**J**) Malpighiaceous trichomes. (**K**) Dioecious sexual system. (**L**) Arboreal and shrubby habit.

**Table 1 biology-12-00173-t001:** Descriptive statistics of the separate and combined DNA datasets used in the phylogenetic analyses.

	ITS	ETS	*trn*L-F	*trn*T-L	*pet*D	cpDNA Combined	nDNA Combined	All Markers Combined
Number of terminals/species	170/86	90/61	174/85	65/53	41/30	178/80	177/81	223/85
Aligned sequence length	750	394	1053	593	1195	2841	1144	3985
Missing data (%)	8.3	4.5	4.5	3.3	16.3	49.8	24.8	54.5
Model	GTR+I+G	GTR+I+G	TVMIV+G	TPM+I+G	GTR+G	-	-	-

## Data Availability

The DNA sequence datasets generated for this study can be found in the NCBI GenBank website. Data on vouchers used in the phylogenetic analyses are included as Appendix A.

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
