# Peer review of "Systematics of Ditaxinae and Related Lineages within the Subfamily Acalyphoideae (Euphorbiaceae) Based on Molecular Phylogenetics"

_biology, 2023, doi:10.3390/biology12020173_

Round 1

Reviewer 1 Report

The article is interesting for the journal, is well presented, and brings novelties to the systematics of the Euphorbiaceae family. It follows the journal's recommendations in terms of format and fits perfectly into its scope.

In this work, the authors examine the phylogenetic relationships between the genera included in the subtribe Ditaxinae (Euphorbiaceae). The complexity of the Euphorbiaceae family (one of the most diversified of the angiosperms) makes these works necessary since many groups lack a solid phylogenetic and systematic framework. To clarify the relationships between the genera included in this subtribe, the authors infer molecular phylogenies using nuclear (ETS, ITS) and plastidial (petD, trnLF, trnTL) DNA sequences and including a large sampling of taxa. With their results, they combine the genus Ditaxis into Argythamnia and update the Ditaxineae at the tribe level. They also establish and describe the tribe Caperonieae and transfer the genus Philyra to the tribe Adelieae. Finally, they propose a new taxonomic treatment.

Specific corrections:

Line 158 à correct H2O to H2O

Figure 1 à this image is difficult to read but the one included in the supplementary material is good.

Lines 59 to 65 à letters are smaller than the rest of the text.

Lines 421 to 423 à the same issue as in lines 59 to 65.

Line 444 à the same issue as in lines 59 to 65.

References are double-numbered.

Line 215 à Jestrow et al. 2012 has to be referenced as [29].

Line 384-385 à In this sentence the citation of Figures 1 & 2 could be confusing because it seems to be two figures cited by Jestrow. I propose to rephrase it: “We propose to circumscribe Philyra within tribe Adelieae (Figures 1 & 2), as suggested by Jestrow [31,53]”.

Review the abbreviation for Müll. Arg. through the text, sometimes the two dots appear out of place.

Some recommendations for the authors:

1. Include a table, preferably in the body text but it could be added in the supplementary material, with all the changes proposed in this paper and the previous classifications. They incorporate this information in Figure 1, but I think that this table could be good for readers.

2. Since cpDNA and nrDNA data reflect different evolutionary histories and their data sets may result in different topologies, I think that it could be interesting to include the phylogenetic trees in which they combine plastid markers and nuclear markers and leave at Supplementary material the remaining inferred phylogenies (all separate markers). There are slight differences in the placement of Philyra and its relation to Adelia. This could help readers to follow the results and discussion.

3. I recommend the authors include the combinations proposed in this paper (Ditaxis grazielae). It would make it highly relevant because here is where the changes are proposed.

Author Response

Dear reviewer,

We greatly appreciate your valuable contributions to the improvement of our manuscript. We accepted all the suggestions included in your review summary. The changes suggested by you and the other reviewer are highlighted in the revised version of the manuscript in blue font. There were a few minor changes that we did not accept. Below we provide a detailed response to all your comments, queries and suggestions.

Please let us know if you have further questions or changes.

Sincerely,

Josimar Külkamp

Reviewer 2 Report

Overall this paper is scientifically sound and presents significant results related to the systematics of part of Euphorbiaceae subfamily Acalyphoideae. The combination of excellent sampling, sufficient DNA sequence selection,  sound phylogenetic analysis, and morphological characters support the authors' conclusions. I have made numerous comments on the manuscript itself. Many simply relate to grammar, punctuation, or word choice. However, there are several that relate to content, and which the authors should address. In addition, I have the following comments.

1. The title is too broad. The work relates only to a small part of subfamily Acalyphoideae, so it's deceptive to have the title say "Systematics of Acalyphoideae...." A more appropriate title would be something like "Systematics of Ditaxinae and related genera (Euphorbiaceae subfamily Acalyphoideae), an approach...."

2. Because several of the genera begin with the same letter, I recommend spelling out the names of all the genera in Figure 1 and the supplementary figures. There is plenty of space to do so.

3. It seemed very strange to first encounter divergence times in the discussion of Chiropetalum, and then to refer to the supplementary material to learn the methods and results. I understand that this part of the paper is largely peripheral, but it still seems that it would be better to mention it in the introduction and include it in the methods and results sections. It's also not clear how divergence time itself supports recognizing a genus without more context, especially some comparison with the ages of other genera. I recommend either expanding this part or deleting it altogether.

4. Please provide more explicit justification for your decision to merge Argythamnia and Ditaxis rather than treating them as three or two genera. See my comment on the manuscript itself.

5. As currently written, the basionym citations for the new tribes are insufficient for the tribes to be validly published. Please review Art. 41.5. Also, new combinations at the rank above genus do not include the basionym author in parentheses (Art. 49.2). Please also note the correct spelling for the tribe Ditaxeae.

Also note that some corrections are needed in the Supplementary material S2. Because I cannot upload more than one file, I will list them here.

1. Change "obteined" to "obtained" throughout.

2. Check the taxon names throughout; some are garbled (e.g., Arlucayanay83 in S1) or misspelled (e.g., shiedeanum should be schiedeanum).

3. At least use the same generic abbreviations in the supplementary figures as in Fig. 1, or preferable spell out the generic names so they are clear.

4. Plastid, not plastidial.

5. In Fig. S10, the names with clade 1 and clade 2 should be Ditaxis, not Argythamnia.

Author Response

Dear reviewer,

We greatly appreciate your valuable contributions to the improvement of our manuscript. We would like to emphasize that we accepted all the suggestions pointed out directly in the text (those you marked on the pdf file) in addition to those included in your review summary. The changes suggested by you and the other reviewer are highlighted in the revised version of the manuscript in blue font. There were a few minor changes that we did not accept. Below we provide a detailed response to all your comments, queries and suggestions. Please let us know if you have further questions or suggestions.

Sincerely,

Josimar Külkamp
